# On Blockchain Integration with Supply Chain: Overview on Data Transparency

Houssein Hellani [1,2,*], Layth Sliman [2,3], Abed Ellatif Samhat [4] and Ernesto Exposito [1]

1    LIUPPA, Université de Pau et des Pays de l'Adour, E2S UPPA, Liuppa, 64600 Anglet, France; ernesto.exposito@univ-pau.fr
2    EFREI Engineering School-Paris, AllianSTIC, 94800 Villejuif, France; layth.sliman@efrei.fr
3    IBISC, Université Paris-Saclay, 91020 Evry-Courcouronnes, France
4    Faculty of Engineering-CRSI, Lebanese University, Hadat 1003, Lebanon; samhat@ul.edu.lb
*    Correspondence: hussein.hellani@hotmail.com

**Abstract:** Data transparency is essential in the modern supply chain to improve trust and boost collaboration among partners. In this context, Blockchain is a promising technology to provide full transparency across the entire supply chain. However, Blockchain was originally designed to provide full transparency and uncontrolled data access. This leads many market actors to avoid Blockchain as they fear for their confidentiality. In this paper, we highlight the requirements and challenges of supply chain transparency. We then investigate a set of supply chain projects that tackle data transparency issues by utilizing Blockchain in their core platform in different manners. Furthermore, we analyze the projects' techniques and the tools utilized to customize transparency. As a result of the projects' analyses, we identified that further enhancements are needed to set a balance between the data transparency and process opacity required by different partners, to ensure the confidentiality of their processes and to control access to sensitive data.

**Keywords:** DLT; Blockchain; supply chain; IoT; smart contract





## 1. Introduction

Supply chain transparency is emerging as a fundamental feature of business continuity and high product quality. Effective collaboration among the different stakeholders requires a supply chain with a high degree of transparency [1]. In fact, transparency enables the different participants in the supply chain to obtain full visibility in terms of the data, services, and products being introduced and exchanged. Different works in the literature have used the terms transparency and traceability to describe this feature. However, these two terms designate two related yet different features [2]. Data transparency is defined [3] as the ability to easily access and work with data, independently of where they are located or what application has created them. On the other hand, traceability in a supply chain is described by ISO 9000:2005 as the ability to identify a product at any stage in the supply chain. It is also defined as a process of tracking the products' provenance and their inputs from the start phase to the end-use. From our perspective, transparency in the supply chain refers to the disclosure of information to trading partners, shareholders, customers, consumers, and regulatory bodies. It captures high-level information along the supply chain, such as product components, suppliers' names, the different locations involved, and associated certificates. Referring to the previous definitions, we conclude that traceability is a prerequisite to transparency realization. Traceability provides opportunities to determine supply chain efficiency, meet regulatory requirements, and verify sustainability claims. To this end, many modern supply chain projects use a different technical solution to achieve traceability, and hence achieve a high level of transparency.

In addition, trust is an essential requirement in a transparent supply chain. Research studies [4–6] show that mistrust among the partners of a supply chain is a significant issue,

which hinders collaboration [7,8]. The supply chain is composed of independent partners, each of which represents a standalone centralized system. Consequently, data transparency may be compromised by a lack of trust among the partners and require more solid trust to be developed [9,10]. Furthermore, consumers may request details concerning the products, including manufacturing origin, quality of service, and proof of safety. Thus, building trust is achieved by enabling transparency along the chain so that individuals and companies can trace their products back to their origin. This can be achieved using Internet of Things (IoT) technology [11,12]. IoT technology is used to deliver the collected data over the network to enhance supply chain performance and traceability. However, the supply chain becomes constrained by additional data loads within the partners' independent systems.

In order to overcome the issues related to trust, Blockchain and, more generally, Distributed Ledger Technology (DLT), is a good candidate which enables the full transparency of data records. It enhances trust between partners through a cryptographic-based, peer-to-peer decentralized platform that underlies a supply chain [13]. Using the Blockchain platform for the supply chain eliminates the ambiguity behind the group of independent databases of traditional supply chain systems, as all records are stored within the ledger on every stakeholder system. Furthermore, Blockchain is immutable against the altering or removal of any records without leaving traces. This is because all partners have a copy of the same updated ledger that leads to a clear vision over the ledger contents. According to many studies [14–16] that have surveyed the critical aspects of implementing Blockchain solutions, Blockchain is a convenient tool to overcome trust and collaboration issues in a supply chain. It is called the "truth machine" [17], and discourages companies from any misconduct. Moreover, many proofs-of-concept (POCs) or piloting schemes have been developed in recent years using technology to study supply chains for traceability and transparency purposes [16]. Data transparency is a built-in feature of Blockchain due to the decentralized nature of the platform. In this context, the ability to control data privacy or opacity within the public Blockchain is questionable, while stakeholders in the supply chain have sensitive data that should not be disclosed to the public. Supply chain projects go far beyond the offered transparency and add their enhancement preferences in addition to the current Blockchain transparency feature. Despite its high importance in building the modern supply chain, there are no comprehensive studies that categorize and analyze the Blockchain-based supply chain's data transparency. To the best of our knowledge, a few of the intended projects have shed light on this topic. The contributions of this paper are:

- We surveyed the existing DLT-based supply chain projects leveling data transparency;
- We investigated the techniques utilized in the data transparency enhancement process;
- We shed light on the importance of transparency and borders between transparency and opacity through access control to successfully integrate Blockchain into a supply chain;
- We highlighted the smart contract and IoT technology roles in achieving controllable data transparency, and call for further investments.

The rest of this paper is structured as follows: Section 2 presents the methodology adopted in this survey. The supply chain transparency requirements and challenges are shown in Section 3. The benefits of using DLT technology in supply chain systems, including transparency, are discussed in Section 4. Section 5 summarizes the existing DLT techniques for supply chain transparency. In Section 6, different projects that employ the DLT in their supply chain are analyzed, with a specific focus on their transparency and traceability features. A discussion is detailed in Section 7 and Section 8 concludes the paper.

## 2. Methodology

This paper contributes to the provision of knowledge about the supply chain's data transparency. The increasing necessity of obtainable transparency in the supply chain encourages investigations in this area of research. The integration of the supply chain with Blockchain and the great evolution towards decentralization are considered by many studies. However, research works that investigate data transparency in the supply chain

are limited. Table 1 enlists the existing surveys and studies that discuss the transparency topic and its effectiveness in the supply chain.

**Table 1.** Existing studies related to Blockchain-based supply chain data transparency.

| Sources | Roles |
|---|---|
| [18] | Elaborates the role of NGO's brand collaboration in enhancing the supply chain transparency |
| [19] | Develops system architecture to integrate Blockchain, IoT, and data analytics to provide sustainable products |
| [20] | Studies the relevance of supply chain transparency to supply chain sustainability governance |
| [21] | Conducts the adoption of Blockchain for supply chain transparency |
| [22] | Reviews transparency/traceability of Blockchain-based supply chain in the literature |
| [23] | Develops smart contracts to directly directly the supply chain transparency |
| [24] | Proposes multi-chain platform to enhance cross-border e-commerce supply chain traceability |

The integration of Blockchain with the supply chain is a relatively new approach. This new approach was adopted to attain immense product traceability and sustainability among companies and individuals. This paper sheds light on the efficiency of Blockchain data transparency and traceability within the supply chain, and illustrates the evolution of data transparency in the modern decentralized system compared to its presence in the traditional centralized system. It also investigates the major transparency challenges encountered in any supply chain and highlights the significance of utilizing the new decentralized platforms.

The research points are summarized in the following questions:

Q1: What are the challenges related to data transparency in the supply chain?

Q2: What are the influences of Blockchain over data transparency in the supply chain?

Q3: What are the existing DLT techniques to achieve transparency in the supply chain?

Q4: Which supply chains are integrated with DLT? How do they tackle transparency?

Q5: What are the logistic obstacles that would affect the achievement of controllable transparency?

Q6: What additional measures should be taken to enhance supply chain transparency?

Our study addresses these questions by investigating the related surveys and studies mentioned in Table 1. However, additional work is required to address the research questions and present the techniques and frameworks which are accountable for building the supply chain transparency based on DLT and IoT technologies. To make progress in this direction, we first considered the papers that entirely or partially tackle supply chain transparency, hosted in scientific databases such as ACM, Elsevier, IEEE, etc. Secondly, we studied the 24 available projects and pilots that decentralize their supply chain and highlighted the used techniques. We also explored many available white papers related to the investigated projects. Some of the projects mentioned in this paper have no detailed technical background. However, we shed light on their contributions and targets.

The novelty of this paper lies in the following aspects: Elaborating on answers to the above questions; showing the different techniques that help the researchers and developers adopt the expedient supply chain platform, regarding their utility; directing future researches towards the topic of transparency access control. This paper can be considered as guide to data transparency researchers. It helps them to build and develop additional transparency techniques for their projects because it analyzes and illustrates the existing up-to-date projects, as well as their innovative techniques, which are utilized to achieve the required supply chain transparency.

## 3. Supply Chain Transparency Challenges and Processes

### 3.1. Data Transparency Challenges

At present, the global supply chain consists of a complex network of stakeholders across industries to coordinate collaborative tasks and achieve mutual agreements. Figure 1 depicts the significant supply chain challenges: centralized systems, lack of transparency, scalability, challenges to IoT integration and the upcoming technologies.

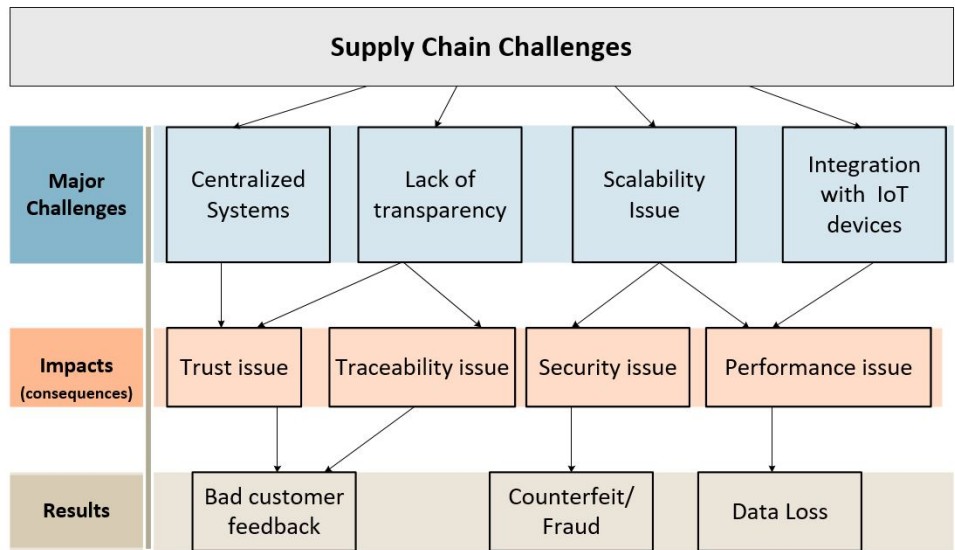

**Figure 1.** Technical Supply Chain Challenges.

The existing centralized supply chain systems struggle unproductively to provide a portion of the vital requirements using workarounds and trusted third parties [25], in addition to the great integration of new technologies. Such independent databases have trust issues resulting in negative customer feedback and dissatisfaction. In addition, there is no reliable shared information within most of the supply chain, and that is the main transparency issue with a centralized system. Lack of transparency leads to traceability and trust issues, in addition to negative feedback from customers. Furthermore, scalability is a major problem when the product travels across many geographical regions. It comes with crucial documents such as ISO certificates, invoices, customs, letters, proofs, etc., and requires hundreds of communications among stakeholders. A study showed that 200 communication processes are necessary to achieve a single product delivery [26]. Hence, the occurrence of scalability challenges leads to security and performance issues. Therefore, these cause counterfeits in the data of the intended products and, in many cases, data loss. This may cause trust issues among partners and result in customer dissatisfaction. Moreover, the current network infrastructure cannot exploit IoT's full potential and manage/analyze the massive incoming data well within the centralized circumstances [25]. In this way, a considerable portion of the IoT power is dismissed. Currently, there are unreliable frameworks and infrastructures designed to connect billions of heterogeneous and disparate IoT devices and their associated services, as well as data aggregation and data analysis [27].

### 3.2. Data Transparency Motivation

The supply chain has encountered enormous changes over time due to the high demands for supply chain transparency and traceability. These demands represent the main motivation for creating transparent systems. For example, when consumers increasingly wonder about where and how their clothes are made, or just how sustainable their potential new electric vehicle might be, given the raw materials required to make it, transparency in global supply chains becomes a notable issue, whicg needs to be addressed [28]. Moreover,

the supply chain has become increasingly involved in the diversity of partners, products, and customer desires. Recently, the challenges have ranged from the heterogeneity of the systems to the additional technological requirements. Thus, besides the above challenges, the motivations behind supply chain transparency are the following:

- Independent Database: The current supply chain infrastructure is a group of centralized-based systems where each stakeholder represents a centralized system which belongs to one or more supply chains. These systems rely heavily on centralized, often disparate, and standalone information management platforms [15]. The group of databases involved in the production process is distributed, heterogeneous, and autonomous [29]. Therefore, data interchange between different databases is inflexible, due to the hard-coded nature of different data standards; Walmart and Cisco are two obvious examples [30]. Practically, the organizations' tendency to use their platform and control their data would limit collaboration.

- Lack of cooperation: The supply chain challenges are mainly related to the heterogeneity of the involved stakeholders, different data forms and lack of communications among the involved systems. Collaborative relationships determine how data are shared between companies, and project them to the underlying business processes. Collaboration is an opportunity for modern businesses to optimize their relationships with their trading partners. However, achieving collaboration poses complex contests between the supply chain actors. In this setting, there is a broad spectrum of collaborative initiatives, disparate standards for communication, and various levels of trading partner competencies and business processes [31].

- Data Loss: The widespread of IoT adoption triggers profound changes in global manufacturing [32]. The IoT systems are usually heterogeneous and categorized under different administrative domains [33]. IoT technology ameliorates the production progress and provides a high level of control, but it charges servers and peripheral devices with a high data volume [34]. The current network infrastructure cannot exploit the full IoT potential and cannot thoroughly manage/analyze the massive incoming data well within the centralized circumstances. Investing in IoT technologies in the current supply chain infrastructure surcharges these traditional systems with high data load, so part of the information is considered lost [25]. Moreover, most valuable products are controlled and tagged electronically; these tags may be cleared/replaced during the transition between stakeholders without leaving traces, leading to trust and security concerns. The probability of data alteration is very high through the current supply chain processes [35,36], where data loss and fraud are likely to happen in many situations.

- Product Complexities: Today's products and services' dispersed natures require their supply chains to be adequately visible to avoid obscurity and provide transparency and traceability features [16]. However, many manufacturers and sellers encounter information insufficiency, and therefore fail to provide customers with the required information due to lack of transparency. Hence, the supply chain complexity is increasingly evolving, as the diversity of the products and requirements requires the integration of many multi-tier supply chains. The availability of high transparency achieves a multi-tier supply chain and manages the different supply chain network parties Thus, the centralized system's uncontrolled informational data lead to massive counterfeit, massive trade losses and bad business reputations.

### 3.3. Security Challenges

Many different security factors challenge the supply chain in a way that may hinder the whole production processes.

- IoT technology proliferation: It is involved in most supply chain chain productions and processing tasks. Their proliferation will exceed half-trillion within the next few years [37]. The IoT devices communicate among themselves, servers and storages, producing massive transaction numbers along with supply chain production lines,

leading to to numerous security challenges to protect the devices and the sensitive data from any leakage or attack.

- Data opacity requirement: Usually, the manufacturing processes are accompanied by several private aspects, including proper planning, recipes, manufacturing intelligence, etc. Data privacy is one of the apparent concerns of the supply chain areas. Therefore, all systems may face data breaches, theft, leaks, unauthorized access, eavesdropping, etc. Accordingly, data opacity must be maintained by all the stakeholders that form a supply chain. By definition, a system is opaque if an external observer is unable to infer a "secret" about the system behavior [38]. Consequently, the decentralized platform that manage the supply chain should consider the opacity requirements.

### 3.4. Supply Chain Policy Enforcement

To achieve the supply chain transparency target as planned, a deep understanding of the intended goods and their requirements is desired. Furthermore, it is required to map suppliers and processes and clarify information gaps. Unfortunately, there is unclear description of the transparency processes that illustrate the road-map of a supply chain project in the literature. In [39] a practical guide to defining, understanding, and building supply chain transparency in a global economy is presented. It is done by: identifying and visualizing risk, using transparency levers to close information gaps, managing, and finally monitoring. In the below, we set the transparency processes for a supply chain to be well employed within better conditions:

- Self-identification: this is the first step that should be settled for a supply chain to identify the environment's overall components, including suppliers and sub-suppliers. Consequently, they should define each component issue and the common intersection among the partners. Accordingly, the risks and the goals are determined afterward, based on the different regulations and rules of the internal/external stakeholders in addition to the common factor impact on the business success.
- Collect information: Collecting data about the production processes, goods, gaps and others, practically on sites, is the most sensitive step. Nowadays, companies increasingly require more data from their suppliers. Collecting accurate data, in this step, is significantly crucial and impacts directly the overall supply chain transparency.
- Expose: the decisions are taken in the last step where the company has a complete picture of the supply chain. The decision takes into account meeting the relevant regulatory requirements and internal/external stakeholders' demands. Furthermore, the company should clarify how the information is disclosed.

### 4. DLT-Based SUPPLY Chain Benefits

Blockchain is a good candidate to address the above supply chain issues. It is a technology used for storing and transmitting data, which guarantees its integrity and transparency. A DLT platform that works without a trusted third party contains the history of the data exchanged in the network. This secure database is replicated in all the network nodes. Blockchain contains a chained set of blocks; each block contains a list of transactions and some other specific data. It is a fully decentralized P2P (peer-to-peer) system that guarantees trust between the non-trusted partners [40]. The main features of Blockchains are their decentralization, shared ledger, tamper evidence, tamper resistance, record-keeping, immutability, distributed trust, multiple-party consensus and independent validation [41].

There are four main types of Blockchain as shown in Table 2: Public Blockchain (permissionless) is available for anyone, such as bitcoin. Public Blockchain (permissioned) is open to anyone for data reading but restricted for data input, consortium Blockchain is a network of predefined organizations, and private permissioned is limited to one enterprise. The Blockchain serves the supply chain against many limitations and improves its functionalities in reference to the features mentioned below:

- Decentralization: The distributed ledger of a Blockchain-based supply chain empowers the involved partners to detect any deterioration of information. Thus, Blockchain tackles data corruption, hacking, or crashing issues in the centralized and independent systems and increases the information validity [42]. Moreover, this decentralized system can be inexpensively implemented among the suppliers [43];
- Trust: transparency is the main consequence of the distributed ledger technology where participants have a complete vision of the current contemporary information. Furthermore, privacy and anonymity are enabled because of the cryptography system [44]. Thus, it is unnecessary to evaluate the trustworthiness of the participants in the network with a decentralized supply chain. Evaluating trust between participants is due to the Blockchain's underlying technology, which guarantees the integrity of data records even in the presence of fraudulent nodes. Therefore, participants recognize that the information is accurate because each involved party has the same data, which cannot be altered or deleted. For this reason, resolving trust issues is discussed as one of the main arguments of the implementation [45];
- Automation: Blockchain applications are mainly based on smart contracts to verify the execution of transactions between two or more parties relying on predefined rules and conditions. The smart contract is a self-executed program or script, which is located on Blockchain ledger [46]. It executes its code once triggered, either from a participant node or from another smart contract. Then, it broadcasts the content to all network nodes for validation and updates the ledgers accordingly in case the contractual terms agree. This automated process reduces the apprehension behind the traditional contract of a supply chain where there is no need for human intervention and trusted intermediaries [47].

**Table 2.** Blockchain types, source: [48].

| | | READ | Write | Commit | Example |
|---|---|---|---|---|---|
| Blockchain Ledger Types | Open | Public Permission | Open to anyone | Anyone | Anyone | Bitcoin, Ethereum |
| | | Public permissionned | Open to anyone | Authorized participants | All or subset of authorized participants | Supply chain platforms viewable by public |
| | Closed | Consortium | Restricted to an authorized set of participants | Authorized participants | All or subset of authorized participants | Multiple banks or chain of restaurants operating a shared ledger |
| | | Private permissioned | Fully private or restricted to a limited set of authorized nodes | Network operator only | Network operator only | Enterprise ledger shared among head office and branches |

## 5. Existing DLT Techniques for the Supply Chain Transparency

The emerged DLT-based supply chain proposals involve several techniques and solutions on top of their decentralized platforms. These techniques are presented in this section to be reused solely or unitedly by researchers for their under-construction supply chain systems. Then, in Section 6, we explore existing projects, and we show how they use the existing techniques related to data transparency and traceability to achieve the modern supply chain requirements. Below are the most valuable techniques:

### 5.1. Blockchain Core Improvement

The peer-to-peer system surcharges IoT devices with computing tasks and high storage demands. Many nodes are designed and prepared with top resources for computing and transaction validation in the mining and cryptocurrency field, which is not the case with heterogeneous IoT devices. Modifying some critical Blockchain characteristics boosts the integration of IoT devices with the DLT technology. Block size, creation time, and consensus algorithm are the areas where altering and adjusting the Blockchain in accordance with IoT requirements occur. In terms of data transparency, the enhancements of the transaction

format have a significant impact compared to the others. The transaction in its current status lacks many essential details. Thus, developing the transaction format to include some real identity, references, blockinfo, etc., will increase the challenges related to the data transparency.

### 5.2. Smart Contract

Smart contracts are automated contracts embedded in the Blockchain, which make the entire process decentralized. Upon the deployment of the smart contracts, it is almost impossible to alter its code. Smart contract is a recent term that is widely used to refer to low-level code scripts running on an Ethereum Blockchain Smart contracts have recently attracted interest due to their importance in business applications and supply chains. In addition to the smart contract's role in ensuring the contract progress, it is also considered an excellent tool to enhance the data supply chain transparency [23]. When a smart contract is executed, all the intended parties within the supply chain are informed of the result and, therefore, they can trace and monitor their products, which increases the transparency level.

### 5.3. Involvement of IoT Device

At present, IoT technology represents one of the core elements of any modern supply chain. It has two main functions: capturing data from media and transmitting them to their destination. Moreover, IoT devices play a central role in ensuring the success of the supply chain's product traceability. Exerting additional efforts has an extensive role in improving the IoT functionalities, to make them suitable for Blockchain-based manufacturers' requirements. Considering the IoT side by the DLT system and its application is the best practice. Some projects [49–52] mentioned in this study have contributed to the IoT improvements in terms of detecting data accurately, to improve the visibility of the data. The transparency is enhanced, and the traceability will be more efficient with the IoT technology involved in the DLT-based supply chain [15].

### 5.4. Merkle Tree Tool

The Merkle tree structure [40] is a binary tree with an associated value for each node, where each one is the hash of its children. These data structure trees are created by repeatedly hashing pairs of nodes until only one hash is left. The last node of a tree (leaf) is a hash of transactional data where other nodes are hash of their previous hashes. This allows any party to quickly verify the validity of data in a branch or leaf using the tree's root hash. Blockchain, and especially Bitcoin and Ethereum, fundamentally use it. Merkle tree has three main advantages over transactional processing. First, it guarantees the integrity and validity of the data. Second, it consumes less memory and CPU resources, as the proofs are computationally easy. Third, Merkle processing requires tiny data to be sent over the network and stored on disks. We involved the Merkle tree in the list of transparency techniques due to its importance in traceability within a supply chain.

### 5.5. Zero-Knowledge Proof

Zero-knowledge proof (ZKP) [53] is an encryption scheme where service providers do not recognize the data stored on their servers. The Prover can prove that a specific statement is true to the other verifier party without revealing any additional information. It can be used in messaging, authentication, storage protection, and for any other sensitive information. ZKP can also be integrated with Blockchain and, more specifically, with the private Blockchain, so that whatever the number of Blockchain nodes, ZKP adds a robust layer of security to the data ledger. Integrating ZKP with Blockchain encourages the supply chain to increase its transparency level, while their data confidentiality is preserved [54].

## 6. Existing DLT-Based Supply Chain Solutions

The supply chain is essential to all businesses. Therefore, the integration of DLT into legacy industries and different stakeholders aims to revolutionize the global supply chain with decentralization features, smart contracts, and IoT technology. Currently, many DLT-based projects seek to acquire trust, transparency, collaboration, and cost/time-saving throughout their innovative DLT platforms. The authors in [55] listed all the 105 DLT-based projects integrated with IoT since 2008 and categorized them into four types without revealing their technical sides. Most of them consist of API interfaces run on Ethereum, the well-known global Blockchain, but they do not have explicit technical references or detailed publications. This section sheds light on the DLT-based supply chain projects and displays their technical parts while focusing on data transparency and traceability. Below are the intended projects that employ DLT in their supply chain:

- Dietrich et al. [23]: proposes an academic framework designed to tackle supply chain transparency by employing a new smart contract approach. The authors achieve their goals by following three steps. In the first step, the framework identifies and enlists all the partners involved in the manufacturing process. The first step is not an easy mission in a complex supply chain, but it is necessary to simplify the manufactured product's concrete process affiliation and composition. This framework assumes that each asset should have a unique identifier. Accordingly, a link is established between each physical asset and the Block-chain platform by generating smart contract's unique identification numbers. These numbers are called virtual identities or Hash'ID, where each one is mapped to a unique physical asset. Hash IDs can also refer to licenses, certificates, or other types of non-physical assets. They are attached to a bar-code form such as Radio Frequency Identification (RFID) or Quick Response (QR) code to link these numbers to the Blockchain. The proposal introduces two types of players in the framework, the supplier and the Certifier. The certifier's role is to assign certification to suppliers in order to create the Hash'IDs. Depending on the supply chain's characteristics, the Certifier's role can be taken over by the manufacturer and other independent organizations. In the second step, they logically attach all the supply chain processes logically into the Blockchain platform through the smart contract. Furthermore, the last step makes the final decision based on a multiple smart contract recorded on the immutable Blockchain ledger.

- Ambrosus [56] is an industrial project utilized to track products throughout their circulation in the market. It is a Blockchain-based supply chain dedicated mainly to protecting and controlling pharmaceutical and food quality. This platform solution is mainly composed of a customized version of Ethereum Blockchain integrated with a data storage solution named the interplanetary file system (IPFS). To avoid the high cost of running transactions on the central Ethereum platform, Ambrosus develops its independent customized version of Ethereum. Moreover, Ambrosus does not rely on Ethereum storage to store the supply chain data, as it is limited in capacity. Instead, it uses IPFS as the primary storage for their large transactions to provide scalability and high throughput for the clients. Ambrosus has advanced sensitive sensors to detect and analyze particular cases related to food and medicament. Ambrosus takes advantage of the Merkle tree in their transactional processes, since it is based on hash cryptography. With this tree algorithm, users can quickly find its data and filter out the wrong inputs. Two types of smart contract are introduced: the requirement smart contract describes quality standards directly compared to items inside the measurements' smart contract, while the measurement smart contract stores the list of ambrosus-certified devices, the root hash of the Merkle tree, and the collected attributes throughout the supply chain to note the variation in composition quality, if any. The Merkle tree data are uploaded periodically to the leading Ethereum network to assist users with further visibility and quickly achieve the tracking process. Ambrosus uses IoT hardware and sensors to tag products, therefore allowing goods

to be tracked through the supply chain and assuring the complete integrity of data comprehensions and transparency.

- Modum [52] is a supply chain for monitoring solutions, which controls the distribution of immense volumes of sensitive goods, especially pharmaceutical ones. It comprises the Ethereum network, the API applications, and a specific sensor called a modum temperature logger. Modum architecture is constituted of front-end and back-end phases. The back-end is composed of an Ethereum network, smart contracts, and a specific server, connected directly to the external users. The front-end is composed of sensors and mobile applications, connected to the HTTPS server in the back-end via REST API and JSON. The logger or SensorTag is the top added value used to measure the shipments' environmental conditions. In detail, each logger has a unique MAC address represented in the QR code, and each shipment has its unique QR, named "track and trace". Both QR codes should be scanned with the user's mobile applications and sent to the server. Once the combination of QR codes is received, the server broadcasts the smart contract and then stores the smart contract ID on the sensor. The client scans the "track and trace" code and requests the sensor's temperature measurements via Bluetooth low energy (BLE). The smart contract receives the data for verification purposes and sends a report back to the client's mobile. When using the smart contract, data authenticity is confirmed at every ownership alternation. The results of the evaluation are then immutably stored in as a proof-of-existence. Using the Modum technique, the data transparency is well-tackled, and there is no need to physically verify the product content.
- OriginTrail [51] is a supply chain solution composed of a combination of off-chain and Blockchain networks. It implements the off-chain network on DLT-based nodes within a new type of decentralized environment. The Blockchain platform runs on different nodes and interacts with the non-DLT network. The off-chain network, known as OriginTrail Decentralized Network (ODN), comprises data and network layers. Thus, the architecture is the stakeholders' applications, the non- decentralized ODN, and the platform. OriginTrail uses Zero-Knowledge encryption to prove private information without revealing it. Moreover, the smart contract is involved in the different off-chain nodes to guarantee the execution of a set of predefined conditions. The aim of using this platform is to store the data fingerprint, ensure the integrity and transparency of records and provide an immutable supply chain system.
- Vechain [57] is a supply chain solution composed of vechain supply chain projects and a vechainthor Blockchain-based platform. Vechainthor is an enhanced version of Blockchain, forked and improved based on the Ethereum codebase. The enhancements cover the transaction format in many directions. The new transaction format includes four fields: independent ID, DependsOn, Blockref, and Expiration. Thereby, the application deal with a single transaction instead of a bundle of transactions. Blockref provides more information about the previous, current, and next blocks. Furthermore, it provides information on the transaction creation time. This will be helpful for financial purposes in case of an acceptance delay, for example. An expiration is added to the transaction to avoid stacking for a long time. Multi-task transaction: a transaction is composed of many small transactions to address complex business payments. Vechain connects the technologies RFID, QR codes, Near-Field Communication (NFC), and bar codes to Blockchain to tag the items with a universally readable identity. The combination of new transaction fields and the IoT technology allows for the accurate tracing of the origin of items and prevents counterfeiting, since Blockchain records cannot be alternated or duplicated.
- Waltonchain [49] is a Blockchain platform designed to track the RFID-based transactions with multiple supply chain partners. It comprises a central network called the parent chain and many other sub-chains networks, which are connected and mined to the parent chain. A sub-chain works independently after being created and registered in the parent chain. The parent chain ledger contains only information related to

sub-chain details, while each sub-chain has its ledger. At any time, a sub-chain can be created and registered to the parent one. The parent chain runs independently of sub-chains, so it does not store the different sub-chains' data. The smart contract is the foundation of waltonchain that builds and maintains the underlying logic platform. Furthermore, it develops an RFID IC tag to be suitable for Blockchain applications. The supply chain sustainability and transparency are managed through the default Blockchain Ethereum platform and the IoT enhancements that help collect accurate data.

- Devery [58] is an open-source protocol based on the Ethereum network, used to build applications for verification purposes where retailers can assign unique signatures to their products. These signatures are stored on Ethereum and used to verify a product throughout the application queries. Devery protocol consists of three data structures, which interact with Ethereum through DeveryRegistry.sol and DeveryTrust.sol smart contracts. The data structure is based on the registration of a product public key with an application's unique identifier. The hash of the product information determines each product's identifier and allows for a lookup via a check method. Devery uses the Entry Verification Engine (EVE) token for payments and charges. The application consumers must pay the application host for the product verification service using Bokky's Token Teleportation Service' (BTTS), which does not permit consumers to directly deal with EVE or gas tokens. This protocol allows for supply chain verification throughout the Blockchain smart contracts without directly interacting with the decentralized environment. This protocol enrolls the transparency over applications by referring to the default Blockchain features.

- CoC [59] refers to "Chain on Blockchain", a supply chain management platform based on hybrid Blockchain to mainly tackle the trust issue of multiple entities. In general, in an authorized network, some nodes are promoted for block creation and validation. CoC distinguishes between the record submission and block-building using a hybrid model to organize the underlying distributed ledger. Submitted records are limited to users, third-party users, and supporting entities only, while building blocks are opened to the public users, named helpers. CoC invented an approach to build a distributed ledger called "Two-Step Block Construction" within their hybrid platform. Step 1 is the generation of reservation blocks by users, and step 2 is to generate data blocks. In step 1, a user submits a request to reserve predictive blocks. The request includes requester information, the fee the user wants to pay for the block, the helper's identity and who creates it, and other essential information. The helpers have to reach a consensus to reserve the block. In step 2, the user uses their reserved block(s) to send data to the ledger. There is no proof-of-work computation effort for the reserved block in this step, since helpers already validate it at the reservation time. The two-step block reservation does not reduce the latency for the overall performance. It provides a mechanism to shift the latency as long as a user has enough reservation. The latency of adding a new supply chain record can be very low. In short, CoC proposes a new DLT hybrid mechanism, but in terms of transparency, it relies on the embedded Blockchain features only.

- Shipchain [60] is a fully integrated system of the entire supply chain that enables tracking shipments from the moment of leaving the factory to the final receiver doorstep. It is based on Ethereum Blockchain using side-chain and smart contract techniques. Records are stored on the Ethereum network, while the side chains can be used by organizations to store and validate their data on their own for cost-saving. Thereby, data is fully decentralized, and located in either Ethereum main ledger or side chain ledger where no mediator is engaged. Additionally, Shipchain contains a web platform that enables shippers to connect directly to carriers without passing the traditional brokerage model. Although the smart contracts run on the Ethereum network, they can be duplicated and operating on Ethereum forks (ShipChain protocol) and used by side-chains for cost-saving. As a result, each shipment has a unique smart contract

that gives shippers more visibility across their supply chain, allows carriers to communicate quickly reducing delays and miscommunications, and achieves the overall required transparency.

- Aqua-Chain [61] is a traceable system for the water supply chain management based on Blockchain and can be implemented by either Ethereum or hyperledger Blockchains. The data transparency is guaranteed, since IoT devices collect data along the supply chain and store them within the Blockchain ledger. Therefore, Aqua-chain software is adapted to provide full traceability to their customers under the classical notion "from-supplier-to-buyer." It is composed of a layered architecture that relies on Blockchain and IoT to achieve traceability. Aqua-chain can be integrated into existing traditional systems such as ERP and CRM. The front-end layer is composed of API REST applications that can easily be integrated with other software. The middle layer is called the controller. It is responsible for transforming the high-level function call into a low-level Blockchain call, and vice versa. Aqua-Chain enables integrating IoT and DLT technologies, and creating transparent, fault-tolerance, immutable and auditable records that can be used for the water traceability system.

- Tael (WaBI) [50] is a decentralized application that permits creation a secure link between Physical and Digital assets through RFID labels with anti-copy functionality. It is independently installed on the user mobile so that they can authenticate their product via mobile app. The user is incentivized through the mining represented by the scanning process, where they perform proof-of-purchase for every scan. Wabi refers to the Walimai organization and supports the "Walimai label," which is applied at a designated point of origin along the supply chain. The registered products under the "Walimai system" consume WaBI tokens. The Walimai label technique provides transparency by being attached uniquely and securely to the product throughout its journey to the consumer. Linking physical products with a unique encrypted code allows the consumer to scan its unique code via mobile applications to check the physical product against its digital state.

- TE-FOOD [62] represents one ecosystem that integrates all food partners (farmer, producer, transporter, and consumer) equally, for successful farm-to-table food traceability, to fight against food frauds and mistrustful supply chains. TE-FOOD introduces a utility token called TFD, Blockchain protocol, smart contracts, plastic security seals, and RFID identification tools. Two types of Blockchain are involved in the progress: the public Ethereum used for the payment process with a TFD token and a private Blockchain to store the transactional data. Accordingly, supply chain companies must have two wallet types: a wallet on the Ethereum network, which can be accessed directly or through the TE-FOOD mobile app, and the transaction wallet on the private network, which TE-FOOD can access The public Blockchain mediates the consumers and the suppliers' private Blockchains. The consumer buys traceability services via the public Blockchain to track its product from different suppliers. TE-FOOD deposits then the purchased transactions to the suppliers' wallets. The contribution of this project can be seen as "traceability as a service", where achieving transparency is an investment for suppliers;

- Cargox [63] is a decentralized solution involved in global transportation, which is implemented on the Ethereum Blockchain. It tackles the bill of lading documents and avoids the supply chain's logistic trading. Users interact through the API of the dApp to create their smart bill of lading. They can either consume cargo token "CXO" directly or utilize the USD/CXO conversion mechanism. Since dApp, based on the public Ethereum with a smart contract enabled, participants can benefit from the DLT transparency feature. Thus, exporters, carriers, importers, or any other parties involved in the transportation, can use their mobiles to manage the shipments. Cargox emerges a full dApp, which allows the carrier to initiate the smart bill from scratch. The carrier sends the bill to the exporter's address once the latter pay the shipping costs, and transfers the bill's ownership to the importer after paying the price of goods.

The importer claims ownership of the goods at the destination port, using the smart contract technology embedded with the dApp. The Cargox dApp empowers users to create smart bills quickly. Besides this, only the involved parties can read these documents, which enforces transparency along with global trade;

- CargoCoin [64] is a decentralized supply chain platform based on the Ethereum network, which aims to encapsulate all types of transport cargoes into a single platform, then connect it to the traders of goods. The services' platform and the smart contracts are both utilized within the platform to achieve this objective. The platform offers a range of communication channels between the partners involved in the supply chain process, providing a method of sending, receiving, rejecting, approving, or signing documentation. CargoCoin introduces smart utilities represented by smart contracts and payments to provide a transparency advantage to the EcoSystem participants and save time and money. The participants, including shipper, carrier and consignee, interact and set terms and conditions using the same decentralized platform;

- ProductChain [65] is a consortium Blockchain, introduced to enhance the traceability of the food supply chain (FSC), taking into account the speeding up the transaction rate into less than one second. It mainly relies on a three-tiered sharding architecture to improve scalability and ensure data availability to consumers. It also introduces the Access Control List (ACL) to limit access to competitive partners, collectively managed by consortium members, and provides read and write access. In addition to its improvement over scalability, it introduces transaction vocabulary to store different types of information and interactions, which encompass all FSC processes. The transaction vocabulary can link the final product to multiple raw ingredients relevant to a broad range of SCs. Productchain enhancements provide data transparency so that a user can quickly trace it back to specific key ingredients and a consortium-governed access control, which guarantees that no participant controls the Blockchain;

- Bext360 [66] is a supply chain platform used to enhance the global food commodities and provide full transparency from farmer to table. It is a software as a service (SaaS), which integrates Blockchain and sustainability measurements to provide a traceable fingerprint from manufacturers to consumers. It runs a RESTful API that allows retailers and wholesalers to insert the technology into their websites, point-of-sale systems, or supply chain management tools. The SaaS platform allows each stakeholder to track food products independently throughout each phase of their supply chain and enhances its overall transparency;

- FarmaTrust [67] provides a robust cloud-based platform to track pharmaceuticals through a supply chain that links digital systems to pharmaceuticals moving in the physical world. It is based on Ethereum Blockchain with a POA consensus algorithm to enhance the scalability. The FarmaTrust platform named "Zoi" is shared among the global community, including suppliers, logistics and shipping companies, wholesalers, distributors, pharmacies, and hospitals. This global network uses the FamraTrust platform to ensure data transparency to ensure that medicines and related medical products are genuine. As a final step toward complete transparency, the consumer is allowed to use the FarmaTrust mobile app to verify the product's authenticity via a QR code scanner;

- BlockGrain [68] is an agriculture supply chain that allows farmers, brokers, and companies to track the path of grains from the place of harvest to consumer destination. It is a decentralized platform using Ethereum Blockchain for the agriculture supply chain. It is structured into three layers: public Blockchain, Private Blockchain, and applications. The main data, smart contracts, and transaction Agri tokens are stored on the public Blockchain, while buyers use the private Blockchain to reduce the transaction costs and waiting times associated with the public. Both Blockchains are managed through the BlockGrain Platform (Applications Layer). BlockGrain automates the delivery process end-to-end. The data are collected and stamped

at each point, along with the product circulation. Using this process, BlockGrain increases visibility and improves transparency;

- ZERO defects [69] is a platform specialized in creating digital twins through an innovative investigative framework for traceability purposes, detecting and mitigating defects early during the production. It is a data acyclic graph (DAG)-based platform managed by the IOTA Foundation with the collaboration of Pickert (an ISO-certified company). DAG is an alternative DLT technology of Blockchain. It has good adaptation with high incoming data loads, and has nominal transaction fees and computation. As a decentralized application, the IOTA platform enriches Zero defects with full transparency and visibility for its board's immutable records. With Zero defects, each product is identified using its unique serial number, which is immutably stored and accessible in the IOTA Tangle;

- Blockverify [70] is an anti-counterfeit Blockchain-based solution for luxury supply chain items. Blockverify tracks each product, which has a unique special tag along the supply chain, where the customer itself determines the transparency level. Blockverify consists of public Bitcoin and an authorized Blockchain to successively store public and private information within the public and private ledgers;

- Chronicled [71] is an industrial supply chain project to gain trust and automation among companies by integrating the Mediledger network. The MediLedger Network combines a secure peer-to-peer messaging network and a decentralized Blockchain network as the ultimate transparent bridge between trading partners. It uses smart tags and the Chronicled App to track the physical products and link them to Blockchain using "identity inlays and tamper-evident cryptographic seals". A Smart Tag is a cryptographically secured chip containing details about the physical good and linked with a private key, which guarantees additional data transparency among all involved parties;

- Everledger [72] comnis a Blockchain platform specializing in protecting the integrity of worthy products, such as diamonds, based on a hybrid Blockchain system. It uses hyperledger as a private Blockchain and Ethereum as a public Blockchain to focus its target on the immutability of the diamond transaction history rather than the system scalability. Everledger combines artificial intelligence (AI), nanotechnology, and IoT to create a digital twin of every single product. This technique provides a secure and permanent digital record of an asset origin, characteristics and ownership. This technologies combination enhances transparency along customer supply chains to enable traceability in a secure, immutable and private platform;

- Fr8 [73] is a supply chain network that aims to modernize logistics with an improved solution for the industry in general, leveraging Blockchain technology at its core. It is based on coupling shipment tracking IDs, RFIDs, and other documentation to create meaningful relationships among multiple datapoints. The Fr8 protocol is composed of five layers. The transport document layer contains the data and metadata of a shipment. The permission & ID Layer manages data integrity and permissions. The interface Layer exchanges data between the document layer and the service Layer. The service Layer connects the Fr8 Protocol with applications. The application Layer works with services and interface layers to display the data. To ensure transparency, Fr8 relies heavily on the Blockchain principle as a single source of truth for shipment data. All of the involved stakeholders will have unprecedented visibility into shipments and their associated data;

- NextPakk [74] is a delivery service that tackles the last mile issues based on Stellar's Blockchain due to its speed and scale. It allows customers to schedule delivery within an hour at home when the package arrives. Furthermore, NextPakk uses Blockchain technology to track packages while protecting customer identity and ensuring a punctual delivery This adds transparency to the delivered goods, where customers can instantly track their packages online. Nextpakk involves Blockchain in elaborating

the entire last mile, so that the consumers can track the driver and obtain complete transparent information on their packages' exact arrival time.

## 7. Discussion

Production stakeholders seek collaboration to optimize the supply chain processes and maintain robust relationships with trading partners. Collaboration among different independent systems challenges the supply chain partners, as there is a broad range of collaborative initiatives, disparate communications, and numerous levels of trading competencies and business processes. A collaborative supply chain requires suppliers and sub-suppliers to share data within a fully transparent environmental media to entirely realize the benefits of collaborative business. Before Blockchain, one of the well-known methods used to achieve transparency is called one step up, one step down. Many supply chains use this principle for traceability purposes [75]. This principle requires each supplier to share their information between the other adjacent ones. In other words, it is a chain of shared information where each supplier receives enough information on the incoming commodity, and then they thoroughly deliver the complete information to all the involved suppliers. It is a neighboring process for actors to share the information among themselves. However, this method is limited to two strides of visibility and, therefore, the transparency is not fully achieved. Furthermore, FarmaTrust [67] finds that technologies, such as holographic tamper-proof labels and unique serial numbers, are not sufficiently effective within the current centralized supply chains. In addition, the challenges mentioned in Section 3 necessitate the intervention of a decentralized Blockchain, coinciding with the development of many other technologies, such as IoT and others. Certainly, Blockchain is a quantum leap toward a new supply chain concept. The new supply chain data are collected differently and added to the decentralized chronological system, which is immutable, anti-counterfeit, transparent, and trusted. Nevertheless, as transparency represents the core of a successful supply chain, what else can be done to the standard traits of any Blockchain network?

We have previously mentioned various solutions targeting the modern supply chain improvements integrated with Blockchain, which enriches the system with trust, transparency, and traceability. These projects integrate the Blockchain within their platforms to overcome the trust issue at the first stage and obtain the other DLT systems' added values. Besides the excellent facilities of Blockchain, the listed techniques in Section 5 are used/introduced by several projects, and they are implemented in various ways to achieve more flexibility in data transparency and traceability. Table 3 shows that these projects' techniques are used to enhance product traceability and data transparency. Referring to the above classification, the most utilized techniques involve IoT device and the smart contract. Most of the projects use these two techniques differently based on their requirements. IoT technology is often used for tracking and tracing items using technologies, such as QR codes, smart tags, RFID tags, NEC, and mobile applications. Moreover, there are some additional IoT devices which are essentially constructed for supply chain transparency purposes. Some projects utilize the smart contract as it was programmed with Blockchain, such as Ethereum. Furthermore, the smart contract is developed to ensure the transparency of off-chain networks outside the Blockchain environment. Some smart contract enhancement tools are represented by assigning different roles or defining multiple smart contracts within the same project, such as setting the standard requirements and measurements of Ambrosus [56]. The Merkle tree algorithm is a technique used to quickly and accurately filter out the wrong data inputs using the crypto-hash functions. The zero-knowledge proof is used to protect sensitive data and enhance transparency. At the level of Blockchain core improvements, one of the techniques used is changing the Blockchain transaction format to include additional fields, which enhance transparency and facilitate traceability.

**Table 3.** Transparency techniques of supply chain DLT-based Projects.

| Project Name | Transparency Technique | Tool |
| --- | --- | --- |
| Ambrosus [56] | Merkle Tree Algorithm | Hash-based data structure |
| | Smart contract | Measurement and requirement smart contracts |
| Modum [52] | IoT device involvement | "track and trace" QR code |
| | IoT device involvement | Modum temperature logger |
| | Smart contract | Normal utilization |
| Vechain [57] | Blockchain core improvement | Block transaction format (ID, DependsOn, Blockref) |
| | Smart contract | Normal utilization |
| Chronicled [71] | IoT device involvement | Smart tag (cryptographically secured chip) |
| WaltonChain [49] | IoT device involvement | RFID tag IC |
| | Smart contract | Manage parent chain and sub-chains contracts |
| Devery | Smart contract | Smart contracts for registration and verification |
| OriginTrail [51] | Smart contract | Off-chain utilization |
| | Zero-Knowledge Proof | Sensitive data protection |
| Cargocoin [64] | Smart contract | Normal utilization |
| Bext360 [66] | Smart contract | Normal utilization |
| Shipchain [60] | Smart contract | Normal utilization |
| WABI [50] | IoT device involvement | RFID cryptographically secured chip |
| TE-Food [62] | IoT device involvement | Plastic security seals (1D/2D barcodes) |
| | Smart contract | Normal utilization |
| FarmaTrust [67] | Smart contract | Normal utilization |
| | IoT device involvement | QR code scanner via mobile SMS/voice label code on traditional mobile |
| ProductChain [65] | IoT device involvement | Transaction vocabulary |
| BlockGrain [68] | Smart contract | Public/private Blockchain managment |
| Zero defects [69] | Blockchain core improvement | IOTA DLT platform |
| Everledger [72] | IoT device involvement | Inteliggent Labelelling: RFID, NFC |
| FR8 [73] | IoT device involvement | Combines RFID, ID, product information |

The DLT integration with the supply chain radically solves the data transparency and provides end-to-end traceability, with clear visibility of all the platform components. Moreover, some of these projects employ extra efforts and propose an additional layer of transparency. They target data traceability, by introducing mechanisms with added values over the current Blockchain features. Different enhanced tracking methods are deployed, ranging from involving new sensors to tags and tracers, as shown in Table 3. Vechain and Ambrosus are notable projects, which employ different methods. Ambrosus takes advantage of the Merkle tree in their transactional processes, since it is based on hash cryptography. With this tree algorithm, users can immediately find their data and filter out the wrong inputs. The tree algorithm can also be used with other IoT devices or mobile scanner applications that distinguish between massive Blockchain records. In other terms,

it enhances the tracing function of the supply chain and speeds up the transparency process. Vechain is moving towards the improvement in the core of Blockchain for additional service refinements. This modifies Blockchain's transaction format by defining new fields: ID, DependsOn, Blockref, and Expiration for each transaction. From a logical standpoint, the Vechain proposal can be commonly used within any DLT-based supply chain. The new fields of vechain can be classified under tracking parameters that can be used with any DLT platforms without challenging their functions. These parameters improve the data transparency and aid in the perfect traceability achievement.

### 7.1. IoT for Transparency Enhancement

The importance of IoT integration with Blockchain to enhance supply chain transparency and traceability rises with IoT technology development. The IoT devices' prominent features are represented by collecting accurate data, quick adaptation, and always-on availability services compared to traditional manual methods. Under the current central structure, IoT experiences the difficulty of achieving a genuine cooperation because the relevant parties of such cooperation often belong to different suppliers with complex or uncertain trust relationships. Therefore, the collaboration of the current IoT devices can only be employed in a trusted environment. As a technology that offers the service of trust, Blockchain can ensure the authenticity of data on the network. IoT ensures the true effectiveness of information when uploaded from the original source. The combination of IoT with Blockchain opens up a road of innovation, with unlimited possibilities. It can be used primarily to track the history of different goods. Thus, IoT technology is essential for new business systems. Furthermore, the IoT helps to establish a harmonious relationship between Blockchain and the world, as IoT devices are the physical interfaces that collect data. In addition, the IoT technology can reduce the disturbing factors from the source to ensure the data's actual effectiveness. Mainly there are five IoT techniques involved in the industrial field [34]: RFID, wireless sensor network (WSN), middleware, cloud computing, and IoT software. In contrast to human abilities, IoT techniques assist producers in collecting data accurately, such as perceiving temperature variation, calculating the elapsed time, and the color degree [76].

Many projects enforce the transparency of the supply chain by introducing the IoT technology within their projects, as illustrated in Table 4. Each project utilizes this technology differently. Waltonchain is directly related to the inventor of RFID technology. It introduces an enhanced RFID version for a Blockchain-based supply chain that provides tamper-resistance, reliability, anti-counterfeiting, and traceability to the business system. Thus, in addition to Blockchain features, the Waltonchain project includes an RFID tag IC and reader IC, appropriate for Blockchain applications. The ICs are characterized by integrating an elliptic curve and decryption acceleration module based on the existing RFID technology and a communication interface protocol for Blockchain applications. Waltonchain solves major IoT problems in Blockchain-based applications. It exempts tags from data storage and limits its responsibility to signature verification. Tags automatically generate random public keys private keys to ensure that the IoT application tag is unique, authentic, and tamper-resistant. Thereby, tags can reduce the amount of information stored to solve overload with large amounts of data in IoT applications. Moreover, tags solve the problem of slow encryption and decryption in asymmetric encryption technology. Modum fabricates another RFID IoT device called Modum logger temperature. The logger is an IoT temperature sensor designed for medical products that do not require active cooling during transport. During the shipment process, the monitored temperature is stored in the logger memory. Using Bluetooth technology, the shipment can be checked without opening it. The results of each evaluation are stored in a smart contract inside the immutable Blockchain. This combination of IoT, Bluetooth and smart contract demonstrates that drugs have not been exposed to conditions which may compromise their quality and integrity. Vechain upgrades the chip layer of a traditional IoT component by adding personal identification with an asymmetric key algorithm. It generates random IDs of 20 bytes, hashes and transforms

them into In this way, every IoT equipment is defined by a unique ID and asymmetric key. These IDs are managed by smart contracts and permanently stored on the Blockchain. Different technologies can be used to achieve the same goal. Wabi and Everledger are both interested in linking digital and physical assets through IoT and Blockchain. However, they use different IoT tag devices.

**Table 4.** IoT-enabled DLT-based supply chain projects.

| Project | IoT Technology | IoT Role | Technology Base |
|---|---|---|---|
| Modum [52] | Modum temperature logger | Trace drug temperature instantly | Smart contract and BLE |
| WaltonChain [49] | IOT-RU20 (RFID tag IC and reader IC) | Upload data direct to Blockchain and realizes Anti-counterfeit | UHF Android Smart RFID Reader/Writer |
| Vechain [57] | Encrypted chips tag technology development | Monitor and trace | Adds ID and asymmetric keys to IoT devices |
| Wabi [50] | Walimai | Links digital and physical assets through RFID labels | Secure RFID label Authentication is done through mobile consumers |
| Everledger [72] | Intelligent Labeleing | Links digital and physical assets through RFID, NFC, | NFC, RFID beacons, and synthetic DNA |

### 7.2. Smart Contract for Transparency Enhancement

The complex manufacturing networks' structures challenge the supply chain transparency and affect the overall collaborative system. The smart contract can reasonably tackle the transparency gap and organize the collaboration. In addition to its central role in drawing legal contracts among Blockchain members, a smart contract enforces the tracking and monitoring of the content of the intended product's data. Some of the Blockchain-based supply chain projects listed in Table 3 use smart contracts for transparency enhancement purposes. They integrate it differently, based on their infrastructure needs. In [23], a smart contract is used for transparency by proposing a framework that interconnects the smart contracts and the manufacturing supply chain' assets. In this proposal, each asset is assigned a unique identification number by generating a particular smart contract stored on the Blockchain. Therefore, the Blockchain ledger can be seen as a database of timestamps that offers anyone the ability to notice that a certain thing has occurred. Ambrosus involves smart contracts in a novel way by introducing two types of smart contracts: measurement and requirement. All the assets and standards are periodically used in the measurement contracts, and the smart contract requirements determine whether a product continuously meets the standards defined by an interested participant in the network. In this framework, the smart contract is utilized as a new protocol to set quality standards and compared directly to the Measurements Smart Contract items. Vechain uses Ethereum virtual machine (EVM) with additional extensions on the contracts called built-ins. It has six smart contract extensions for further data reliability.

Generally, the complexities of attaining transparency are caused by the stakeholder's incompatibility in rules and conditions, leading to difficulties in reconciling transparency requirements. The partners experience the obstacle of not revealing private data while attaining intended transparency. The smart contract is a trusted tool that plays a significant role in achieving transparency and some data privacy. All regulations, rules, and conditions related to different supply chain partners should be collected at the first step to identify the risks and goals. Hence, a smart contract transcribes/records and stores all the regulations, conditions, and risks on the immutable Blockchain ledger. It is committed to executing the partners' recommendations by literally and intelligently following their predefined code content. The smart contract can then help achieve the planning requirement, and can assist it in reaching its targets successfully. Coupling smart contracts with IoT technology interacts directly with the sensors to ensure precise execution. The registered data represent a source of trust for all partners, since it is recorded on an authentic ledger. Finally, the

decisions to expose data are taken through a dynamic and trusted platform, considering all the supply chain' network complexities.

### 7.3. Transparency Versus Opacity: Access Control

In addition to the above-mentioned technical limitations, there are further obstacles that would affect the ability to achieve full transparency. Full (uncontrolled) transparency goes against the opacity and privacy required by stakeholders that intend to hide sensitive information such as plans, cost, secret ingredients, etc. In a Blockchain-based supply chain, and since its introduction, Blockchain ensures the use of multiple keys for signing transactions so that, each time, a new key is generated and used once. This method protects user privacy on the cryptography level. However, it requires an advanced method to enable parties to customize their transparency and opacity levels based on their requirements and regulations. For example, some sensitive data are shared among partners/companies only for collaboration purposes in a supply chain. Thus, the decentralized platform is requested to protect such data from leakage and provide certain opacity using access control. At this level, questions are raised about the effectiveness of the aforementioned techniques on transparency control. How can we mitigate the gap behind Blockchain data transparency and obtain access control?

In this context, the fully decentralized system (public Blockchain) prevents partners from controlling their data as if it were in the centralized systems. This inability to control opacity leads the partners to prefer the private Blockchain to the public one, knowing that the latter is much more recommended for the global supply chain. Hence, there is a need to accomplish the access control feature within the global Blockchain. Table 5 depicts the techniques' impacts on the transparency access control, their advantages, and limitations. Starting with the public Blockchain, the ZKP promises to preserve their privacy, encouraging them to go public. Other cryptographic proposals would also have a large impact on data privacy, such as homomorphic encryption integrated with Blockchain [77]. These algorithms protect privacy and advance public Blockchain usage. However, they have a medium impact on data transparency and opacity control. The Merkle tree technique facilitates traceability without exerting a significant impact on the control side. Regarding transparency, smart contracts and IoT techniques can significantly provide access control if employed precisely. The recent projects listed above, which investigated smart contract development, ignore the transparency access control and concentrate on their functionality part only.

**Table 5.** Current techniques impacts on transparency access control.

| Techniques | Transparency Access Control Impact | Benefits | Limitations |
| --- | --- | --- | --- |
| Zero-Knowledge-Proof | Medium | Ensure privacy in public Blockchain and encourage merging supply chains | Unable to recover lost user credentials |
| Merkle tree | Medium | Facilitate extract and tracking data | Hash collision and overhead syncing |
| Blockchain core improvement | Low | Facilitate the access control in case of improving transaction format and roles | Have no direct impact unless the improvements become related to data transparency |
| Smart Contract | Very High | Apply conditioning access control and automate the traceability process | Complexity in a scalable environment |
| Involvement of IoT devices | High | Rapid data correlation and facilitate automation | Unable to be managed in a vast centralized system |

To satisfy both transparency and opacity requirements, a hybrid Blockchain is an appropriate solution to regain partners' confidence in the supply chain decentralization. On

the one hand, partners are able to run and store their sensitive data off-chain, thus achieving opacity. On the other hand, partners could publish different data to the Blockchain to ensure transparency. In addition, a hybrid smart contract [78] was recently proposed, which permits the control of off-chain data and lets partners build smart contracts that cover both on-chain and off-chain data.

*7.4. Summary and Open Issue*

As a result of these analyses, the Blockchain offers an attractive embedded transparency feature to the entire supply chain and improves the overall processes. Furthermore, the techniques presented in this paper assist in the achievement of further transparency and support supply chain actors in building their platforms. The current Blockchain-based supply chain solutions lack transparency access control. Thus, restructuring their platform is required. The techniques that could be used are very different: The smart contract technique has the highest impact on transparency access control. The IoT technology integrated with smart contract automates the traceability process to enhance transparency and reduce the overall risks. Besides, other cryptographic techniques, such as the ZKP, encourage enterprises to accord with the public Blockchain while conserving their privacy. This technique may drive the adoption of open supply chain platforms in the future. Any supply chain planning to move into Blockchain could pass through the above policy enforcement steps for the required validation to recognize the best-fit Blockchain type and techniques. After this, it can explore the techniques mentioned above that fit its requirements. Nevertheless, the data transparency topic still takes considerable effort. The aforementioned projects lack transparency standards to manage and organize the data transparency requirements within the new DLT technology. This paper aims not to prompt actors to choose between different projects (like Vechain, Ambrosus, OriginTrail, etc). Instead, it sheds light on the available techniques and drives stakeholders to integrate more techniques in different ways to mitigate the gap behind the transparency concerns of the new introducing DLT-based supply chain. Another goal is to highlight the transparency access control topic and its influence on the decentralized supply chain projects. The analyses highlight the open issue related to inventing more tools to improve transparency in general, and specifically to advance the progress in the transparency/opacity access control.

## 8. Conclusions

This work highlights essential questions related to the Blockchain-based supply chains' data transparency. It considers the transparency challenges, the DLT transparency techniques, and additional measures that can be employed. Accordingly, the existing projects are presented with their adopted techniques. It is noted that some of them implement standard Blockchain features, including transparency and traceability. However, some other projects exploit additional techniques to enhance transparency and satisfy their requirements. We highlight these techniques and analyze them to investigate their impacts on a Blockchain-based supply chain. IoT technology and an advanced smart contract are the most-used techniques to achieve more transparency, as well as what Blockchain provides. Few projects use alternative methods, such as involving cryptographic tools like Merkle tree and zero-knowledge. We conclude that further enhancements are needed to achieve the required data transparency in the supply chain and to control unlimited access to sensitive data according to the opacity requirements.

**Author Contributions:** Conceptualization, investigation, methodology, and analysis: H.H., L.S. and A.E.S.; original draft preparation: H.H.; writing—review and editing, H.H., L.S. and A.E.S.; Supervision: L.S., A.E.S. and E.E. All authors have read and agreed to the published version of the manuscript.

**Funding:** This research received no external funding.

**Institutional Review Board Statement:** Not applicable.

**Informed Consent Statement:** Not applicable.

**Data Availability Statement:** Not applicable.

**Conflicts of Interest:** The authors declare no conflict of interest.

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
