# Peer review of "On Blockchain Integration with Supply Chain: Overview on Data Transparency"

_logistics, 2021_

Round 1

Reviewer 1 Report

Recently,there are many papers about transparency and traceability issues of blockchain. Some closely related works on this topic are missing in this paper, such as: https://doi.org/10.1016/j.ijinfomgt.2019.102059

By comparing with existing literature, the theoretical and practical contributions and values of conducting this research should be discussed.

I suggest to provide an in-depth comparison of these projects. Not only describe and list them in the content and table, but also classify and discuss on the logics behind ideas, technical frameworks, key techniques, advantages and disadvantages, range of applications etc.

The promising directions of the future development of the integration of blockchain and supply chain should also be a good point to discuss, which will significantly improve the value of this paper.

Reviewer 2 Report

This is a generally well-written and well-structured paper with sufficient theoretical background. In this paper, the authors present – “On Blockchain Integration with Supply Chain: Overview on Data Transparency”. The paper seems interesting. In my opinion, it could be published if the following issues are resolved.

  • The abstract does not reflect contribution of the study. Reading the abstract does not help the reviewer understand why this problem so important to address, which methodologies have been developed, used, and/or applied to solve this type of problem.

  • Highlight the novelty aspects of this study. I hardly see the novelty in the contribution. The authors are asked to show their original contribution in a more convincing way. The reported review and the discussions are normative and minimal -there is nothing new here, most of them are referable in existing publications. More comprehensive evaluations / delving deeper into the subject is needed.

  • The authors can consider improving the supply chain section in discussing security aspects. The following article could be helpful:

-Idrees, S.M.; Nowostawski, M.; Jameel, R.; Mourya, A.K. Security Aspects of Blockchain Technology Intended for Industrial Applications. Electronics 202110, 951. https://doi.org/10.3390/electronics1008095

  • Authors can further check for typographical errors.
